# Refugee Students' Psychosocial Well-Being: The Case of a Refugee Hospitality Centre in Greece

Nektaria Palaiologou and Viktoria Prekate *

Language Education for Refugees and Migrants (L.R.M), Humanities Department, Hellenic Open University, 26335 Patras, Greece
* Correspondence: vprekate@gmail.com

**Abstract:** Education can be important for assisting the psychosocial well-being of marginalized communities such as refugees and contributes to the effective processing of feelings and isolation prevention by mitigating the long-term effects of trauma and developing strategies to manage life changes. A small-scale study was conducted on 21 students from a Refugee Hospitality Center in Greece to investigate their psychosocial well-being through questions about their life, daily activities, former and current school life, family relations, feelings about their past and present, and expectations from their new country of residence. The research was conducted through semi-constructed interviews by a specialist research team and certain sociological factors, such as gender, country of origin, and prior school experience, were examined. The results highlight the presence of severe traumatic histories in the lives of many refugee children, the need to escape from their countries of origin, missed school years, the impact of current schooling on their psychological well-being, and the limitations of camp life. Most children reported school experiences to be among their happiest moments, affirming the importance of schooling in helping children build mental health resilience.

**Keywords:** refugee education; child mental health; psychosocial well-being; refugee trauma





## 1. Introduction

The Greek 'refugee crisis' started in 2015 and, since then, the country has received record numbers of refugees, who have arrived mainly via the Mediterranean route on their way to Western and Northern Europe. Greece has become a transit country for about 1.2 million refugees from warring regions outside Europe [1] on their way to other destination countries. Many became 'trapped' in this first reception country longer than expected [2]. As most refugees came in family units or family fragments, Greece became the home of about 45,000 refugee children in 2020 [3]. These children fled their homes due to violence, insecurity, war, and hunger [4].

Refugee camps (formally known as Refugee Hospitality Centers) became the main type of accommodation provision during the refugee crisis. The refugee camp, where the research was conducted, Skaramagas Refugee Hospitality Center, was one of the most populated camps in the country. Located in the outskirts of Athens, the camp's occupancy varied from 1800 to nearly 3000 with an average of about 2500 people during the research period (2019–2020). Most beneficiaries came from Syria, Afghanistan, and Africa, with 40 percent of the population being minors [5]. Children were offered formal education, with compulsory registration in primary and junior secondary education, although no sanctions were imposed for non-enrolment [6]. Formal education was provided through two routes: (a) Reception classes, offered in morning mainstream schools, with a mixture of parallel intensive Greek language courses and attendance in mainstream classes. (b) Structures for the Welcoming and Education of Refugees (known in the Greek education system through the abbreviation 'DYEP'), which were segregated classes, exclusively for

refugee newcomers, hosted within mainstream school premises [7]. These classes accepted refugee students from all nationalities, who were ranked according to initial assessment tests in host language skills [8]. Nonformal education classes were provided by national and international nongovernmental organizations (NGOs) and several evaluations of nonformal class participation affirmed the progress in refugee children's cognitive and academic skills after program completion [9].

Refugee children are prone to psychological aftereffects, including behavioral difficulties [10]. Relocation to another European country, as much as it was desired, introduced a new kind of stress, adding to past trauma [11]. In a synthesis study on psychological distress among refugee children, post-traumatic stress disorder (PTSD) symptoms are found in 19–54% of the sample and depression symptoms in 3–30% [12]. PTSD symptoms are related to the original trauma but can also arise as a secondary result of other stressors [13]. Past traumatic experiences are correlated with children's PTSD, whereas post-migration difficulties and insecurity about asylum are related with refugee children's depression [14]. Exposure to conflict and violence, the loss of family members, and hunger all pose great risks to the mental health of children and their subsequent capacity to adapt to a school environment and to cooperate and progress cognitively [15].

Other difficulties of psychological nature are also recorded in refugee populations, such as anxiety disorders, psychosomatic symptoms, and behavioral difficulties [13]. Several studies have highlighted the combination effects of past trauma and post-war low quality of life [16], aggravating refugee teenagers' depression symptoms [17]. According to another study, the factors contributing to the low quality of life of Syrian refugees include socioeconomic hardship, asylum status, family break up, accommodation difficulties, etc. [18]. Such problems abounded for refugee children at the Refugee Hospitality Center of this study, Skaramagas refugee camp, as the vast majority faced chronic insecurity about asylum status and securing family income.

Up to 77% of young refugees with traumatic history may experience mental health; however, school attendance, socialization, and linguistic ability seem to predict a lessening of these problems over time [19]. Another research study on Syrian refugee children found that 60% had experienced a traumatic incident where they saw someone's life in danger and 45% experienced symptoms of PTSD [20]. Suggestions about addressing trauma and improving refugee children's psychological health include the training of teaching staff to recognize and deal with trauma, maintaining connections with their home country and mother language, being welcomed in their new environment and included in a supportive school system [21]. Schooling plays an important role for the psychosocial health of refugee students, if it provides a safe and inclusive environment, where communication, positive socialization, and a sense of belonging can flourish [22,23].

The age range for the study's subjects encompasses the preadolescent and adolescent years, a period when children form their personality and develop their life views, sense of self, and goals. Adolescence is a very sensitive time during which several domains interact with each other to affect and shape the mental well-being of the young individual. Psychosocial development in adolescence is a result of dynamic systems interactions at the level of individuals, dyads, groups, societies, and cultures [24]. The systems relevant to this study are, for example, the extended family, local school, nonformal education classes available, and friendships. The study's participants share their views and experiences of these interactions. Adolescents' psychosocial development include abstract thinking, introspection, the adoption of values, the emergence of an independent self, development of peer relations, participation in identity groups, and the ability to develop coping strategies [25]. The question remains regarding how refugee teenagers living at camp can achieve these milestones if they are not integrated into societal groups where they can form sustained relationships, if they are not given enough stimuli to develop coping skills, if their only peer group is limited within the camp, and if they are not given opportunities to develop hobbies and abilities. The isolated mental health interventions, often implemented in Western host countries, usually do not include the bigger picture, in which social functioning, structured

daily activities, and community integration are critical in restoring an overall psychosocial well-being [26].

It is the bigger picture that this case study attempts to elicit, in the specific context of refugee teenagers living in a crowded refugee camp in Greece. Education can be a protective factor in reducing the long-term impact of past trauma. Farasaki's research with refugee children at Skaramagas refugee camp [27] found that school attendance had a positive effect on children's psychosocial well-being and cognitive and academic development. Pieloch et al. [28] also found that socialization, feeling accepted and welcomed by the host community, positive schooling, and family communication and connection promote the resilience of refugee children. Marley and Mauki [29] found that a connection with one's own culture, social support, and feelings of safety and belonging are beneficial to refugee children's psychosocial well-being, while refugee students are effectively assisted to adapt to their new environment, if the school is prepared and implements appropriate practices. Some of these protective factors contributing to psychosocial well-being are examined in this research.

This case study is the first in Greece to qualitatively examine the psychosocial well-being of refugee children living in refugee camps. Greece's refugee education policies require children living in refugee camps to enroll directly in local formal schools that native children attend. Even newcomers, children with no Greek speaking skills, are enrolled in local schools, with transportation provided free of charge. This is a uniquely direct approach, and one favorable argument was to promote psychosocial inclusion and enhance the psychological well-being of refugee children through socialization with local peers. The important contribution of this case study is that refugee children had the opportunity to express, in their own words, their views on their past and present life, their expectations for the future, their feelings about 'going to school with Greek children', and the effect it had on them.

## 2. Materials and Methods

The research intended to investigate refugee teenagers' psychosocial well-being in relation to their past experiences and current living conditions. The research was granted permission upon request to the relevant ministry, where the purposes and possible gains of the study were presented. Three specially trained post-graduate students participated in the visits and conducted the interviews with refugee children and teaching staff. Parental permission was required for children to participate in the study. The principles of 'do not harm' were followed, avoiding triggering questions and explicitly offering the option of not replying. The twenty five interview questions were short, open-ended, non-directive, and posed with sensitivity to the children's needs. The sample, as shown in Table 1, consisted of twenty-one teenagers from Syria, Afghanistan, and Iran, aged 10–17, who had been in Greece for 7–72 months. The age range was chosen because of the capacity of children to respond adequately to open-ended questions and the likelihood of having more conscious memories of their past than younger children.

**Table 1.** Sample composition.

|             | Boys | Girls | Total |
|-------------|------|-------|-------|
| Syria       | 4    | 7     | 11    |
| Afghanistan | 1    | 7     | 8     |
| Iran        | 2    |       | 2     |
| Total       | 7    | 14    | 21    |

The interview protocol contained introductory questions, questions about their journey to Greece, about life at camp, and about expectations from their new settlement. Interviews lasted from 7 to 16 min, depending on participants' willingness to talk. An interpreter was present to assist with translation if needed and the interviews were recorded and later transcribed as text. Children were not pressed for answers and treated with sensitivity,

warmth, and encouragement. If, on some occasions, children wished to remain silent, the researcher moved on to the next question. The participants were given the space to construct their own narrative of experiences, while each participant was respected as a unique individual [30]. All answers were protected with confidentiality and data were presented using a code system so that interviewees could not be recognized.

Careful planning and preparation of the data collection was necessary, as access to refugees hosted in Refugee Hospitality Centers required official permission. The research interviews were conducted by the research team between July and September 2020 during six separate visits at Skaramagas Refugee Hospitality Centre (on the 16th, 23rd, 27th, and 29th of July 2020 and 1st and 15th of September 2020). Prior to the interviews, several preparatory visits had been conducted from August 2019 until November 2019 to establish a climate of trust with the students and the camp's staff. Due to the COVID-19 pandemic, access to the camp was further hindered, while refugee students faced additional difficulties regarding their school attendance and social life [31]. The closures stopped the research between November 2019 and May 2020; however, the research team resumed its work with the target population during May–June 2020.

## 3. Results

The quality of life of the refugee children was investigated in three directions: (a) past and present school attendance, (b) daily life at camp, (c) views on their past and present. The three directions were explored through open questions so that children had the opportunity to actively express their story in their own words. For example, in the first direction, students were asked about school attendance in their home country, the attendance of formal school in Greece and/or nonformal classes at camp, and what they liked about the three different settings.

The second direction consisted of questions about how they spent their day at camp. These questions led to a flowing dialogue between the researcher and interviewees, so that participants were free to share about their daily reality. Activities at camp were grouped into categories and the purpose was to discern monotony and boredom or types of activities that promoted children's psychosocial well-being. The third direction concerned children's views about their past and present. Children were asked about life in their home country, what they liked about it, and what they found difficult. Regarding their current situation, they were asked what their happiest moment had been since arriving in Greece and, in an oblique way, what their expectations were about their arrival.

Generally, participants expressed their feelings eloquently, presenting a clear picture of the severity of their circumstances, sometimes through silence. For some questions, it was deemed appropriate to categorize answers according to their country of origin or gender. Interviewed children appeared to become more talkative when asked to 'Describe us how you spend your day here at the camp', as they had the opportunity to talk about their hobbies, friends, and current reality, indirectly affirming children's innate resilience by focusing on the present moment. Some older children seemed to be more fluent in their comments about their past, including negative situations, whereas others were quieter. One question was indirect: 'What do you think children from other countries want to have when they come to a foreign country, for example, Greece?', making it easier for children to project their own expectations and views, without fear of being judged. Children's answers are displayed below, in seven tables, grouped in the three directions:

### 3.1. Direction A: Past and Present Schooling

3.1.1. School Attendance in Country of Origin

Considering that there was a four-year difference between the research and the onset of the refugee crisis and that all participants were between 10 and 17 years of age, most participants were expected to have attended at least 1–2 years of schooling in their country of origin. However, according to Table 2, six children replied that they had no schooling at all in their country of origin, and a considerable number (six) did not wish to talk about it,

an indicator that schooling (or lack thereof) was a difficult topic for them. Twelve children (four boys, eight girls) did not report school attendance in their home country. Five children commented on the reasons for not attending school, that allowed no doubts about the severity of their situation: 'bombing', 'bombs', 'I did one year, then stopped, there was no heating and there were bombs', 'not safe going to school', 'I risked my life every day to go to school'. Perilous journeys to school and schooling interruption due to a lack of safety are trauma-generating factors, exacerbating other highly stressful conditions.

**Table 2.** School attendance in country of origin.

|  | **None** | **1–2 Years** | **3 Years or More** | **No Reply** |
|---|---|---|---|---|
| Syria | 3 | 4 | 2 | 2 |
| Afghanistan | 2 | 1 | 2 | 3 |
| Iran | 1 |  |  | 1 |
| Total | 6 | 5 | 4 | 6 |

3.1.2. Opinions about Current School Attendance

According to Table 3, most children reported positive feelings about formal school attendance, even though formal schooling poses more challenges than nonformal classes at camp, regarding language and integration with the local community. The pandemic also discouraged some parents from sending their children to school outside the camp. Some of the negative comments about formal schools were expressed by a child who discontinued and a child who kept attending but was dissatisfied: 'We were alone', 'We are at the corner of the class, the teacher doesn't care'. These comments indicate that a non-supportive school environment can discourage children from continuous attendance. A mixed comment was: 'Initially it was difficult, but now it's OK, I have made Greek friends'. Some of the positive comments were: 'It was good to know someone cares about your education', 'I like the Greek children', 'I love my teachers and the gym', 'I like the Greek school, 'I like Kyria (teacher), she's very nice'. The onsite nonformal classes run by the Danish Refugee Council (DRC) nongovernmental organization seemed more readily accepted and received the following comments: 'I like writing', 'I like my teachers', 'I like watching movies in class', 'I like Math and English', 'I like the DRC classes a lot, I am grateful'.

**Table 3.** Opinion about current school attendance.

|  | **Positive** | **Negative** | **Mixed** | **Does Not Attend** | **No Reply** |
|---|---|---|---|---|---|
| Formal school | 11 | 1 | 2 | 5 | 2 |
| Non formal onsite classes | 16 | 1 | 0 | 1 | 3 |

*3.2. Direction B: Current Daily Life*

3.2.1. Daily Life at Camp

An important indicator for psychosocial well-being is the number of activities in children's daily life at camp. According to Table 4, the average number of activities for boys was higher than for girls, although it still remained low for both genders. It is interesting to note that no activities of recreation were mentioned, as well as hardly any common activities with parents. A reason for this is that life at camp could be characterized as limited since the Refugee Hospitality Center was remotely located from the city and transportation was difficult for beneficiaries. The cost of recreation activities, as well as language barriers, are also obstacles when venturing to the outside world. Only one boy replied that he used to go shopping with his father. Some characteristic comments were: 'I get bored when we don't have lessons', 'After classes I stay at home and sit and wait for the night to sleep, it's boring', 'We sit at home, it's very hot outside', 'I like Greece, but not life at camp'.

**Table 4.** Life at camp and daily activities.

| | Play Sport | Swim | Play Games with Friends | Read/ Study | Draw/ Listen to Music/Sing | House/TV/ Help Parents | Nothing/ 'I Get Bored' | No Reply | Average Number of Activities |
|---|---|---|---|---|---|---|---|---|---|
| Boys | 5 | 1 | 6 | 1 | 1 | 1 | 0 | 1 | 2.5 |
| Girls | 2 | 2 | 3 | 2 | 1 | 2 | 4 | 2 | 1.7 |

All children lived with their parents and siblings, mostly in large families. To the question whether they liked life in Greece, thirteen children said they liked Greece and would like to stay. Six children stated that they did not wish to stay, because of relatives in other European countries. Two children said they had mixed feelings. Some of the comments about the host country were: 'I like Greece and Greek people very much', 'I like Greece, but I want to go to Germany because my brother is there', 'I like Greece, but people are not always helpful', 'I like Greece, but not life at camp', 'I like Greece, but it is too expensive'. Many children reported that they liked the sea and the sun. Overall, children responded positively about their new host country.

3.2.2. Communication with Relatives in Country of Origin

It is worth noting that, in Table 5, most children from Syria did not communicate with family in their home country, mostly because there was no family there left. Three Syrian children who reported communicating with family abroad did so with Syrian relatives in Europe (for example, Switzerland, Norway, Germany, Holland). Afghan and Iranian children, on the other hand, seemed to have more ties with their families in their home country.

**Table 5.** Communication with relatives in country of origin.

| | Often | Sometimes | Rarely | Never | No Family There | No Reply |
|---|---|---|---|---|---|---|
| Syria | | 1 | 2 | 3 | 4 | 1 |
| Afghanistan | 1 | 4 | 2 | | | 1 |
| Iran | 2 | | | | | |
| Total | 3 | 5 | 4 | 3 | 4 | 2 |

*3.3. Direction C: Views on Past and Present*

3.3.1. Quality of Life in Country of Origin

Participants were asked: 'What would you like to tell us about your life in your home country? What did you like and what was difficult for you?'. In Table 6, it is seen that eighteen out of twenty-one children said that life in their country of origin was 'Not so good'/'Very difficult' or stated to the researcher that they didn't wish to talk about it. Their response confirmed that most refugee children fled perilous situations that compromised their survival. The children who answered: 'Not so good' or 'Very difficult', on some occasions, provided the reasons for this on their own initiative (for ethical reasons, children were not asked any further by the researcher). Their answers are presented in Table 7, where eleven children stated issues of safety, war, and fighting, with three adding that there were water/food shortages and electricity cuts, while four said that they could not go to school. Some characteristic comments were: 'There was no food', 'I was shaking everyday with bomb sounds', 'I risked my life to go to school every day', 'Initially life was good, but then bombs started, a lot of deaths', 'Every day we did exercise at school for bombs', 'I saw many dead people', 'Police came to my house to take father and uncles', 'There was violence, it was very scary, here it's is safe, I 'm happy here', 'We didn't go out, we stayed at home', 'We left, but my father returned to the house to take something, but then our house was broken and since then, we had no news from our father', as well as 'I was at home, it

wasn't safe, but I had a special feeling I was at home, I don't have this feeling now' and 'My country was a second hell for women and children'. The comments indicate that most children had been exposed to severely traumatic situations before coming to Greece.

**Table 6.** Quality of life in country of origin.

|  | Good | Not So Good | Very Difficult | No Reply |
|---|---|---|---|---|
| Syria | 1 | 2 | 6 | 2 |
| Afghanistan | 1 | 1 | 6 |  |
| Iran | 1 |  |  | 1 |
| Total | 3 | 3 | 12 | 3 |

Thirteen out of fifteen children, who reported that the quality of life in their home countries was 'Not so good' or 'Difficult', explained their reasons for saying so (children commented on their own initiative, without being asked). The reasons are displayed in Table 7:

**Table 7.** Reasons [1] given for 'Not so good'/'Very difficult' quality of life in country of origin.

|  | War/Safety | Food/Water/Electricity Shortages | No School |
|---|---|---|---|
| Syria | 5 | 1 | 1 |
| Afghanistan | 6 | 2 | 3 |
| Iran |  |  |  |
| Total | 11 | 3 | 4 |

[1] Some children gave more than one reason.

3.3.2. Expectations of Refugee Children When Arriving in Greece

Table 8 presents some interesting results regarding children's expectations upon arrival in the host country. Some of the 'Other' comments about the projected expectations upon arrival were: 'To be treated like a human being', 'To find myself, to express myself', 'To fix our lives, to have a good life', but also 'To go to Germany, we have friends there' (this participant had no expectation of Greece as a place of destination, which was also related to the fact that this child had not registered in formal education). It seems that education was a prevalent expectation of children coming to the country. All children who stated 'education' as their expectation attended formal or nonformal classes; moreover, all children who expressed 'making friends' as an expectation indeed managed to make new friends at camp. The children's answers reflect Maslow's hierarchy of needs, placing education at the base of the needs' pyramid base, along with physiological needs (food, shelter, clothing) and social–emotional needs (safety, security, belonging).

**Table 8.** Expectations of refugee children when arriving in Greece.

|  | Food | Education | Play/Toys | Work for Parents | Friends | Safety | House | Other | No Reply |
|---|---|---|---|---|---|---|---|---|---|
| Boys |  | 4 | 2 | 1 | 2 |  | 1 | 2 | 1 |
| Girls | 4 | 4 |  |  | 2 | 2 | 2 | 1 | 3 |
| Total | 4 | 8 | 2 | 1 | 4 | 2 | 3 | 3 | 4 |

One important note about expectations upon arrival is related to the type of journey children went through. All children from Syria replied that they came through Turkey and then, by boat, to Greece. All children from Afghanistan (except two) replied that they crossed Iran, Turkey, and then came to Greece by boat. Two children mentioned three intermediate countries before coming to Greece (Pakistan, Iran, Turkey), highlighting the magnitude of uncertainty in these children's lives, who may have changed five countries of residence by early adolescence. A characteristic comment about the journey to Greece

was: 'Getting in Greece was the hardest thing that I could think: Uh. Uh ( . . . ) it was extremely difficult'. Bearing in mind the hardship of the journey, the final question 'What was your happiest moment since arriving in Greece?' allowed some salient points to emerge, indicating the children's deeper needs and desires (Table 8).

3.3.3. Positive Experiences: Refugee Children's Happiest Moments since Arrival

The answers presented in Table 9 indicate that the most common reply to the question 'What was your happiest moment since arriving in Greece?' was the arrival itself: 'My happiest moment was when we arrived in Greece', 'I am happy here, it's good here', 'I was very happy to be outside of my home country', 'There is no happiest moment, I like everything', 'When we came here, we had no food, no home and people cared about us, it was a good moment', 'The happiest moment was when we arrived here and left the ship'. Children placed value on arrival to Europe, after a long and dangerous journey. Schooling also seemed to play an important role in children's happiness, as some children expressed: 'My happiest moment was I played the violin in my school orchestra', 'When we visited the astronomy museum with my school', 'My best moment is when we sit at school and we write, Kyria (i.e., teacher) is very nice'. There was also the following comment: 'I will be happy only when I go to Germany'.

**Table 9.** Refugee children's happiest moments since arrival.

|  | Moment of Arrival | School Experience | The Sea and the Sun | Other | No Reply |
|---|---|---|---|---|---|
| Boys | 3 | 3 |  | 2 |  |
| Girls | 4 | 2 | 3 | 3 | 1 |
| Total | 7 | 5 | 3 | 5 | 1 |

## 4. Discussion

Although there was negligible differentiation in most replies among boys and girls, it seems that girls had a more limited range of daily activities at camp than boys and fewer outdoor activities. This is to be expected, as many teenage girls tended to spend a significant portion of their time inside the family's container. Girls also appeared to be shier and more often represented in the 'No reply' category. These differences are worth investigating further; for example, to what extent do familial contexts and cultural beliefs affect adolescent girls' lack of social activity and in what ways does staying indoors affect their processing of past trauma.

On the other hand, educational expectations seem to be similar among boys and girls. It is an encouraging sign that cultural norms about girls' education seem to be progressing towards equality, and migration to Europe appears to be a way towards female educational achievement. Five out of eight boys and nine out of thirteen girls registered in formal Greek schools, even though the study was conducted during the pandemic, a period that negatively affected school attendance for disadvantaged children. Refugee students in Greece showed a significant fall in education participation during the pandemic [32]; yet, despite this, rates of positive regard for schooling remained relatively high. This result is in accordance with the emphasis placed by refugee children on advancing their studies to tertiary education and achieving successful careers. Opinions were favorable for nonformal classes too, confirming other findings that the provision of education, can contribute significantly to students' learning, social, emotional and linguistic needs, as well as facilitate connections with the local community [33,34].

Secondly, in terms of ethnicity differences, equal numbers of children from Syria and Afghanistan reported very difficult situations in their countries of origin, providing concrete examples of dire conditions in both countries. Schooling seems to have been hindered in both countries, and all children arrived in Greece by boat from Turkey, a journey which, as they commented, was extremely difficult. Syrian children reported to have much less

communication with relatives in their home country than Afghan children, and this relates to the different sociopolitical developments regarding refugee population movements.

The participants appeared to have a significant degree of trauma in their histories, with adverse effects on their current quality of life. The lack of variety in camp life is notable; yet, children need a balance of activities that develop their cognitive, social, physical, and emotional skills and sustain intrinsic motivation [22]. The main source of such creative occupation in this case was linked to education, confirming multiple literature sources about its role in improving the psychosocial well-being of children.

## 5. Conclusions

A group of twenty-one refugee teenagers from Syria, Afghanistan, and Iran at a large Refugee Hospitality Center in Greece were interviewed about their past and present life conditions to assess factors affecting their psychosocial well-being. The results and comments yielded a picture that is consistent with refugee populations from warring regions and who have histories of traumatic experiences. Refugee children (i) experienced severe obstacles in accessing education in their home countries, sometimes compromising their safety, (ii) had a good rate of attendance in both formal and nonformal education classes in Greece, expressing a positive regard for studying, relationships with teachers and fellow students in these settings; (iii) their daily life at camp seemed to be limited, with a less than average number of daily activities for girls compared to boys; (iv) reported very difficult conditions in their home countries, including war, water/food shortages, electricity cuts, and a lack of schooling, (v) expected their arrival in Greece to fulfil their basic needs, (vi) had varying degrees of communication frequency with relatives in their country of origin, (vii) generally described their happiest moment in Greece to be upon arrival or during an education related-experience. An important insight gained from this study is the value refugee teenagers placed upon education, both as a future objective and a means for current well-being. Although half the children affirmed their intent to move to other European countries, they generally expressed overwhelmingly positive views about education provided in Greece, as well as their wish to be accepted and make friends in their new environment. Schooling is confirmed to be a protective factor for mental health, alleviating the symptoms of past trauma, providing opportunities for socialization with local peers, offering a wider range of enriching daily activities outside the camp, and fulfilling children's expectations about their new country of residence. The realization of the level of danger and trauma these children have been exposed to renders it imperative that all countries of destination provide an inclusive and welcoming school environment, holistic educational programs, and trauma-sensitive educational interventions, which respond to refugees' educational and psychosocial needs equally.

**Author Contributions:** Conceptualization, N.P.; methodology, N.P.; validation, N.P. and V.P.; formal analysis, N.P. and V.P.; investigation, N.P.; resources, N.P. and V.P.; data curation, V.P.; writing—original draft preparation, N.P. and V.P.; writing—review and editing, V.P.; visualization, N.P. and V.P.; supervision, N.P.; project administration, N.P.; funding acquisition, N.P. All authors have read and agreed to the published version of the manuscript.

**Funding:** This research was funded by the HORIZON 2020 Project MICREATE (Migrant Children and Communities in a Transforming Europe), EUROPEAN UNION: 822664.

**Institutional Review Board Statement:** The study was conducted in accordance with the Declaration of Helsinki and approved by the Committee of Research and Ethics at the Hellenic Open University (date of approval: 15 January 2019). The code is RC 8003, 15-1-2019. Article Processing grant: Special invitation by MDPI Guest Editors.

**Informed Consent Statement:** Written informed consent to participate in this study was provided by the participants' legal guardian/next of kin.

**Data Availability Statement:** The datasets presented in this study can be found in online repositories. The names of the repository/repositories and accession number(s) can be found at: All data are registered at the official portal of SODANET The Methodological Protocol was approved by the Relevant Committee of Research and Ethics at HOU. RELEVANT LINK: https://doi.org/10.17903/FK2/JAB51L Research Data on Migration and the Refugee Crisis LINK TO THE COLLECTION: https://sodanet.gr/dataservices/infographics/migration (accessed on 16 January 2023).

**Acknowledgments:** We would like to thank for their assistance the researchers who conducted the field interviews (data collection), Eirini Kyriazi and Marina Sounoglou who were involved at the transcription process, as well as the student team who supported the implementation of the research at camp and all the refugee children involved.

**Conflicts of Interest:** The authors declare that the research was conducted in the absence of any commercial or financial relationships that could be construed as a potential conflict of interest.

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
