# Peer review of "Refugee Students’ Psychosocial Well-Being: The Case of a Refugee Hospitality Centre in Greece"

_societies, doi:10.3390/soc13030078_

Round 1

Reviewer 1 Report

After careful scrutinizing of the content, it has been observed that the paper is good but has a lot of scope of further improvement in the following ways-

1.An extensive review of literature always adds to the quality of the content. Authors are required to add detail about the psycho social systems, how it impacts the feelings and mental well being of  humans. With the addition of the theories on Psycho social systems, the content quality may further be improved.

2. In the Introduction section, a small paragraph must be added by the authors indicating - how the research is unique and contributing to the knowledge reservoir  .

3.In the section materials and methods, authors have described about the sample size, authors are advised to tabulate the data ,as it will bring more clarity.

4.Heading 3.2.1 needs to be changed as it is not justifying the purpose.

5.More clarity is required for the data tabulated in Table No. 6

6.Reference section is very weak. Authors are required to add the references in alphabetical order and that too in the desired format of the journal. Authors must refer to  the guidelines provided by the Journal for this purpose.

7. Authors are required to revisit the whole paper and check the grammatical and spelling errors.

Reviewer 2 Report

The introduction presents a good background for the study but could be improved with a more clearly articulated problem statement and research questions.

Furthermore, the reference list is a little bit weak, author(s) need to strengthen their theoretical framework. A more critical synthesis of the literature (Refugee Students) and the potential construction of a research agenda will also improve the significance and value of the paper.

I feel that this is an interesting paper overall.

Round 2

Reviewer 2 Report

I applaud all the efforts of the author(s) for this research and the revised version of this manuscript. I propose to accept it after minor revisions.

Author Response

Dear Reviewer2

Thank you so much for your comments. Additional changes are noted in grey. Minor corrections were made throughout the text. One reference was deleted and the context of others in the text was improved.  Conclusions were improved by additions. Regarding the IRB Statement, the study was approved by the Committe of Research and Ethics at the Hellenic Open University on 15 January 2019 and the study is conducted according to the Declaration of Helsinki. Details about Funding, Data Availability and Acknowledgements are found in the manuscript with authors' names submitted initially.

If there is anything else we need to add, please let us know.

With best regards,

The authors